# A Latent Class Analysis of Health-Related Quality of Life in Korean Older Adults

**DOI:** 10.3390/ijerph18157874

**Published:** 2021-07-25

**Authors:** Eun-Hi Choi, Mi-Jung Kang, Hyun-Jin Lee, Mi-Soon Yun

**Affiliations:** College of Nursing, Eulji University, Uijeongbu-si 11759, Korea; choieh@eulji.ac.kr (E.-H.C.); yms@eulji.ac.kr (M.-S.Y.)

**Keywords:** elderly, health-related quality of life, latent class analysis

## Abstract

The present study aimed to confirm latent classes in health-related quality of life (HRQOL) in older adults and investigate the characteristics of participants in each class. It aimed to provide basic data to develop interventions for each quality-of-life class by analysing the predictors of each class. Secondary data from a community health survey in G province since 2019 found a total of 41,872 participants. Of them, 9027 were 65 years or older and residing in G Province in 2019, participated in this study. Mplus 8.5 was used to conduct a latent class analysis of five domains of HRQOL. Four latent classes in the HRQOL of older adults, namely, stable type, physical disability type, emotional disability type, and crisis type were found. Certain variables predicted these classes. Based on the findings of the present study, training on functional mobility and balance to prevent falls in older populations and individualised programmes to promote mental health in them should be provided. Moreover, policies should increase medical accessibility and provide social support for older people with low-incomes. Additionally, since physical, psychological, and social health in older adults are inter-connected, a comprehensive care plan is needed to improve their HRQOL.

## 1. Introduction

The South Korean society is aging rapidly. At present, people aged 65 years and older account for 15.7% of the total population, which meets the definition of an aged society. Korea is expected to become a super-aged society by 2025 [1]. With the continued increase of the older population, health-related quality of life (HRQOL) in older adults has become a pivotal issue. HRQOL, which is a subjective evaluation of one’s health [2], can be defined as the quality of life that is influenced by health or diseases [3] and plays an important role in healthy aging [4,5].

According to previous studies, HRQOL is higher in older adults when they have fewer chronic diseases and are more physically active [6,7]. Hypertension and diabetes, particularly, have a negative influence on the quality of life in the older population and influence their physical, psychological, and social health [7]. Moreover, restrictions in functional mobility and balance seen in them may lead to falls [8,9], which increases their dependence in daily life and causes social isolation, all of which decrease quality of life [10]. Therefore, it is necessary to elucidate and evaluate the relationship between HRQOL and physical health, including hypertension, diabetes, and falls.

Meanwhile, HRQOL is conceptually associated with mental health, including dementia, depression, and anxiety [11]. Depression is a common mental health issue among older adults and has a negative influence on quality of life [12]. Depression and anxiety in older adults increase with a reduction in social activities, avoidance of interpersonal relationships, and worsening physical and mental health [13,14]. Moreover, cognitive decline, such as subjective memory deficit, decreased concentration, and decreased calculation abilities, also paves the way for depression and anxiety in them [15,16]. However, some older individuals accept depression as a part of aging, and depression is sometimes not as noticeable as other physical symptoms, which results in a decline in quality of life, leading to a vicious cycle [17]. Therefore, to improve the quality of life of the older population, their mental health should be thoroughly investigated, and ways to promote a positive mental health should be explored.

The quality of life in older adults is also associated with social health, that includes trust in the community [18], active participation in social activities [19], and peer relationships [20]. Choi et al. [21] reported that these factors influence quality of life through the mediation of mental health. Life occurrences such as retirement and moving out of children’s house can cause social isolation in older adults [22], which, in turn, influences their mental health. Therefore, in this cycle of events, it is necessary to understand the factors related to social health in HRQOL. Income can also be included in social health domain, given that Lee and Ko [23] have reported that older adults cope with depressive symptoms differently depending on their income, and that people with low income are more likely to depend on medication.

Previous studies have mostly investigated the factors that influence HRQOL in older adults. However, almost no study has investigated whether latent classes are present in each domain of HRQOL or which factors characterise these classes. Latent class analysis aims to identify subgroups of people with common characteristics in such a way that people within the subgroups have a similar scoring pattern on the measured variables, while the difference in scoring patterns between the subgroups are as different as possible [24]. The present study aimed to confirm latent classes in HRQOL in the older population and investigate the characteristics of participants in each class. By analysing the predictors of each class, we aimed to provide basic data to develop interventions for each quality-of-life class.

The specific objectives of the study are as follows:

First, the participants’ general characteristics, physical health, mental health, and social health are identified. Second, we investigated and named latent classes of participants’ HRQOL. Third, we confirmed the predictors related to general characteristics, physical health, mental health, and social health of each latent class of the participants’ HRQOL.

## 2. Materials and Methods

### 2.1. Design

This descriptive study aimed to confirm latent classes of HRQOL in Koreans over 65 years of age and analyse factors influencing each class. It is also a secondary analysis study utilising the data collected in 2019 from the community health survey of G Province.

### 2.2. Participants and Data Collection

The community health survey used in this study has been conducted every year since 2008 to investigate the current community health, including health-related behaviour and disease contraction, in adults over 19 years of age. The target population is set based on the registered community population as of July of each year and stratified based on administration districts in South Korea and type of residence. The sample was extracted to ensure a consistent extraction probability in proportion to household size, based on the number of households in each type of residence in administration districts. The number of households in administration districts in South Korea selected as samples was calculated for systematic sampling. The present study utilised 2019 data from the community health survey of G Province.

The Korea Centers for Disease Control and Prevention (KCDC) delivered standardized survey performance and survey question guidelines to investigators. All participants gave their informed consent before participating in the survey. For the community health survey used in the present study, trained enumerators visited selected households and conducted face-to-face interviews using laptops with the survey programme, and entered the survey responses directly into the notebook. After the data collection was complete, the households were called to review the data, and the collected data were anonymised. This study was approved by the Institutional Review Board of E University (EUIRB2021-020).

The participants were older people aged 65 years and above residing in G Province in 2019. A total of 41,872 individuals participated in the community health survey of G province in 2019. Of them, 9027 were 65 years or older (21.6%).

### 2.3. Tools

#### 2.3.1. General Characteristics

For the participants’ general characteristics, their age, sex, household type, and low income were analysed. Whether the participants had low income was determined based on whether they responded with yes or no to the question, ‘Is your household eligible for Basic Living Security (note: The government provides economic support to people with lower incomes)?’ Household type was classified into one-person household, married couple household, single parent household, and others. Households with grandparents, single parents, and unmarried children or grandchildren were classified as single-parent households.

#### 2.3.2. Physical Health

For the participants’ physical health, their diagnoses of hypertension, diabetes, and number of falls were analysed. The diagnosis of hypertension was classified based on whether the participants responded with yes or no to the question, ‘Have you ever been diagnosed by a doctor with hypertension?’ The diagnosis of diabetes was classified based on whether the participants answered yes or no to the question ‘Have you ever been diagnosed by a doctor with diabetes?’ For the number of falls, the participants were asked the following question: ‘In the past year, have you slipped, missed your step, fallen on your buttocks, or fallen otherwise? If yes, how many times?’.

#### 2.3.3. Mental Health

Participants’ mental health, depression, discomfort from cognitive decline, and happiness were measured. For depression, the PHQ-9 depression test questionnaire was used. The PHQ-9 was developed by Spitzer et al. and adapted into Korean by Park et al. [25]. The tool consists of nine items, and the total score ranges from 0 to 27. The score for the PHQ-9 is interpreted as follows: 1–4, normal; 5–9, mild depression; 10–14, moderate depression; 15–19, moderately severe depression; and 20–27, severe depression. Participants with scores of 10 or above were asked to seek care. In the present study, scores of 9 and below were coded as no depression, and 10 and above were coded as depression. The Cronbach’s alpha was 0.83 in this study.

For discomfort from cognitive decline, the participants were asked the following three questions: ‘In the past year, how often have you had difficulties with usual household chores due to confusion or memory deficit?’, ‘How often did you require help in daily life due to confusion or memory deficit?’ In the past year, how often did confusion or memory deficits interfere with your work, volunteer activity, or social activity?’ The scores were answered on a 5-point scale from 1 (never) to 5 (always), and the scores for the three questions were averaged.

For happiness, the response of ‘very dissatisfied’ to the question ‘Considering everything, how satisfied are you with your recent life?’ was coded as 1, and ‘very satisfied’ was coded as 10.

#### 2.3.4. Social Health

Participants’ social health, trust in the social environment, degree of contact with peers, and social participation were measured.

For trust in the social environment, participants were asked to respond with yes or no to the following statements: ‘My neighbours can trust one another’, ‘My neighbours help one another during family events’, ‘I am satisfied with the overall safety (natural disaster, car accidents, farming accidents, and crimes) of my neighbourhood’, ‘I am satisfied with the living environment (electricity, water supply, garbage collection, and sports facilities) of my neighbourhood’, and ‘I am satisfied with medical services (public health centres, hospitals and clinics, traditional Korean medicine clinics, and pharmacies) in my neighbourhood’. Answers of ‘yes’ were coded as 1, and ‘no’ were coded as 0. The total score ranged from 0 to 7.

For the degree of contact with peers, the following question was asked: ‘How often do you contact or meet with your relatives (including family), neighbours, and friends?’ Answers were coded as follows: ‘less than once per month’ was scored 1, ‘once per month’, 2, ‘2–3 times per month’, 3, ‘once a week’, 4, ‘2–3 times per week’, 5, and ‘4 times or more per week’, 6. The scores were summed and averaged to obtain a final score between 1 and 6.

Social participation was assessed by asking whether the respondents participated in ‘religious activities’, ‘socialisation’, ‘leisure activities’, and ‘volunteer activities’ regularly at least once per month. The participants responded with yes or no.

#### 2.3.5. Quality of Life (EQ-5D)

The EQ-5D was developed by Rabin and de Charro and standardized by Kim et al. [26]. The EQ-5D consists of five domains: mobility (M), self-care (SC), usual activity (UA), pain/discomfort (PD), and anxiety/depression (AD). Each domain is evaluated as 1 corresponding to ‘not problematic’, 2 corresponding to ‘somewhat problematic’, and 3 corresponding to ‘severely problematic’. In the present study, for latent class analysis, all five domains of HRQOL were reverse scored as 0, corresponding to ‘presence of difficulty’ and 1 corresponding to ‘no difficulty’. Each domain was assigned a score of 0 or 1, with higher scores indicating a higher quality of life. In other words, higher scores in each domain indicated better mobility, self-care, and usual activities. Moreover, they indicated the absence of pain, discomfort, anxiety, or depression.

### 2.4. Data Analysis

The data were analysed using Mplus 8.5 (Muthen & Muthen, Los Angeles, U.S.), and SPSS 26.0 (IBM SPSS Statistics, NYC, U.S.), in the following order according to the study objective.

First, the participants’ general characteristics, physical health, mental health, and social health were analysed in terms of frequency, percentage, mean, and standard deviation.

Second, a latent class analysis was conducted for the five domains of HRQOL. To test the goodness of fit for the selection of latent classes, Akaike information criterion (AIC), Bayesian information criterion (BIC), sample size adjusted BIC (saBIC), Lo–Mendell–Rubin likelihood ratio test (LMR), and bootstrap likelihood ratio test (BLRT) were used. With AIC and BIC, the lowest absolute values indicate good model fitness [27]. While AIC and BIC are influenced by sample size, saBIC is not [28]. LMR and BLRT provide significance probabilities for the class models. The null hypothesis is that the latent class model would be rejected, and the alternative hypothesis is that the latent class model would be accepted. The alternative hypothesis is accepted when the significance probability is ≤0.05.

Third, to confirm factors that predict quality of life in each latent class, a multinomial logistic regression analysis was performed.

## 3. Results

### 3.1. Participants’ General Characteristics and Health Issues

Most patients of this study (35.6%) were aged between 65 and 70 years, with a mean age of 74.07 years. Of the total participants, 56.9% were female, and 6.2% had a low income. In terms of household type, 18.7% lived in one-person households, 45.9% lived in married couple households, 1.9% lived in single-parent households, and 33.4% lived in other households.

In terms of physical health, 57.1% had hypertension, 12.3% had diabetes, and the participants had a total of 0.32 falls in the past year. Regarding mental health, 5.2% had depression, and the mean score for cognitive difficulties was 0.325. The mean happiness score was 6.66.

For social health, the participants scored 5.06 out of 7 for trust in the social environment and 3.46 out of 6 for the degree of contact with peers. Of the total participants, 61.3% had some kind of social participation (Table 1).

### 3.2. Latent Classes for Health-Related Quality of Life

In this study, latent classes for HRQOL were also explored (Table 2). LMR and BLRT were found to be not significantly different in group 6. Additionally, LMR and BLRT were noteworthy in groups 4 and 5, respectively. Since AIC, BIC, and saBIC were the smallest in group 4, it was selected as the final model. In group 4, 57.3% were in class 1, 16.6% in class 2, 10.1% in class 3, and 16.0% in class 4.

The characteristics and names of the latent classes are provided in Table 3 and Figure 1. In class 1, conditional probability for quality of life was 0.95 for mobility, 0.99 for self-care, 1.00 for usual activity, 0.76 for pain and discomfort, and 1.00 for depression and anxiety. The class had a high quality of life and was named the ‘stable’ type. For class 2, the conditional probability for quality of life was 0.04 for mobility, 0.96 for self-care, 0.56 for usual activity, 0.11 for pain and discomfort, and 0.78 for depression and anxiety, respectively; mobility, pain, and discomfort influenced class 2’s quality of life. This class was named the ‘physical disability’ type. In class 3, conditional probability for quality of life was 1.00 for mobility, 0.86 for self-care, 0.98 for usual activity, 0.41 for pain and discomfort, and 0.01 for depression and anxiety. Depression and anxiety influenced the quality of life of class 3. The class was named the ‘emotional disability’ type. For class 4, the conditional probability for quality of life was 0.11 for mobility, 0.30 for self-care, 0.00 for usual activity, 0.08 for pain and discomfort, and 0.26 for depression and anxiety, respectively, and the class had a severely reduced quality of life and was named the ‘crisis’ type.

### 3.3. Predictors of Each Latent Class of Health-Related Quality of Life

According to the latent classes identified above, all factors were entered into a multinomial logistic regression analysis with age and sex as control variables (Table 4).

The regression model was found to be significant (χ^2^ = 3640.915, df = 45, *p* < 0.001). Cox and Snell R2 were 0.341, and Nagelkerke R2 was 0.381. When class 2 was compared to class 1 as the reference group, married couple households compared to other households (OR = 0.84, *p* = 0.019), hypertension (OR = 1.30, *p* < 0.001), number of falls (OR = 1.30, *p* < 0.001), depression (OR = 2.67, *p* < 0.001), discomfort from cognitive decline (OR = 1.47, *p* < 0.001), happiness (OR = 0.87, *p* < 0.010), degree of contact (OR = 0.916, *p* < 0.001), trust in social environment (OR = 1.09, *p* = 0.001), and lack of social activity (OR = 1.39, *p* < 0.001) were found to be significant.

When class 3 was compared to the reference group, one-person households were compared to other households (OR = 1.40, *p* = 0.003), number of falls (OR = 1.26, *p* < 0.001), depression (OR = 4.91, *p* < 0.001), discomfort from cognitive decline (OR = 1.79, *p* < 0.001), and trust in social environment (OR = 0.90, *p* < 0.001) were seen to be significant. When class 4 was compared to the reference group, low income (OR = 2.04, *p* < 0.001), number of falls (OR = 1.56, *p* < 0.001), depression (OR = 10.30, *p* < 0.001), discomfort from cognitive decline (OR = 2.12, *p* < 0.001), happiness (OR = 0.65, *p* < 0.001), degree of contact (OR = 0.91, *p* < 0.001), and lack of social activity (OR = 1.82, *p* < 0.001) were found to be significant.

## 4. Discussion

This study found four latent classes of HRQOL in older adults, namely, stable type (class 1), physical disability type (class 2), emotional disability type (class 3), and crisis type (class 4). Second, the following variables predicted the four classes: low income, household type, hypertension, number of falls, depression, discomfort from cognitive decline, happiness, frequency of contact, trust in social environment, and participation in social activities.

Of the four classes, the largest class was the stable type, with 57.3% of the participants. The stable type was characterised by high quality of life in the health of older adults in all of the following areas: mobility, self-care, usual activity, pain and discomfort, and anxiety and depression. These participants can represent the general characteristics of individuals over 65 years of age and can be used as the reference standard in research on HRQOL in older adults.

The physical disability type, which accounted for 16.7% of the participants, had the lowest quality of life due to physical mobility. Depression and anxiety were lower in this class than in the other two classes, except for the stable type. Even though the class may experience difficulties in self-care due to severe pain and reduced mobility, mental health was maintained relatively well. This agrees with a previous study finding that depression or anxiety lowers self-care and has a negative influence on HRQOL [29]. Therefore, various programmes should be developed and applied to improve depression and anxiety among older adults.

Previous studies have found that chronic diseases influence HRQOL in older adults [7]. In the present study, compared to the stable group, hypertension had a significant influence on HRQOL in the physical disability type, while no clear correlation with diabetes was confirmed. In this study, only data on hypertension, diabetes, and number of falls were analysed in the evaluation of physical health. This is a clear limitation of our study due to the use of secondary data. Various chronic diseases and the presence of their complications are important variables in the association of HRQOL with the physical health of older adults [7,30]. In the future, the association with chronic diseases should be explored further in terms of physical disability. In this study, the physical disability type had trust in the social environment, participated in social activities, and trusted and used social support despite health issues, when compared to other classes. This is thought to improve self-care and influence HRQOL [31,32].

The emotional disability type accounted for 10.1% of all participants, and the quality of life due to depression and anxiety was the lowest. Contrastingly, mobility was highest, and usual activity was also high, like the stable type. This type had an extremely high physical mobility and was able to maintain self-care and usual activities despite severe depression or anxiety. When the emotional disability type was compared to the stable type, significant differences were found in the following variables: number of falls from the physical health domain, happiness, depression, and cognitive decline of the mental health domain, and one-person household in the social health domain. The score for happiness was particularly the lowest in this class, and scores for depression and cognitive decline were also the second lowest, following the crisis type. This finding indicates that this class requires mental health assistance. Although many previous studies have reported that depression in older adults influences their quality of life [33,34], our findings showed that a good physical health can increase quality of life despite low levels of happiness and high levels of depression and anxiety.

Both the emotional disability and crisis types had substantially higher discomfort from cognitive decline, which is a general characteristic. Cognitive decline is associated with negative emotions, such as anxiety and depression, regarding the possible prognosis in terms of progression to dementia [34]. This in turn, has a negative influence on HRQOL [35,36,37].

Lastly, the crisis type accounted for 16.0% of all participants and had extremely low levels of quality of life in every domain. Particularly, the pain-related quality of life and quality of life in usual activities were severely low. Low income is a factor that influences the crisis type. Ordu et al. [38] reported that low economic status in older adults is an important determinant of pain. Low income leads to a lower quality of the surrounding environment for older adults. To improve HRQOL in individuals with low income, it is necessary to improve the safety environment, natural environment, living environment, access to medical services, and physical environment for older adults [39].

The present study found that the physical disability type, emotional disability type, and crisis type all had more falls than the stable type. These participants accounted for more than 1/3 of the target population in this study. Fear of recurrent falls inhibits activities and acts as a major cause of loss of independence, thereby reducing HRQOL in older adults [40]. Functional mobility and balance can help improve the quality of life in the older population by reducing the risk of falls [8], reducing limits on physical and daily activities [41], and maintaining independence.

In terms of social health, the degree of contact with peers, trust in the social environment, and social activities were significant when compared to the stable type. Social support can be considered from the perspective of the degree of contact with peers and perception of older adults [42], and it mediates the relationship between peer relationships and mental health [31]. Lee and Ko reported that older adults were more likely to experience health issues when they had lower levels of contact with peers, had lower levels of trust in the social environment, and participated in social activities less often [23]. Therefore, to improve their HRQOL, the authors suggested promoting social support from the community and strengthening social networks [43]. Kang [44] reported that those who were isolated from the community environment had lower levels of satisfaction with life compared to those who were closer to medical and social support facilities. In other words, because the older population has a reduced scope of social activity, it is important for them to be close to resources that can assist in solving their issues.

The present study has several limitations. First, since this study was designed as a cross-sectional study, it is difficult to estimate causality between variables. Moreover, since this study used secondary data, it was impossible to control all factors related to diseases. Second, although an area with the largest population of older adults in South Korea, which has rural and urban communities, was selected, it is difficult to generalise the findings of this study as social environments differ in each area. Despite these limitations, the present study is significant as it confirmed latent classes of HRQOL in older adults and assessed their predictors.

We would like to suggest the following based on the results.

First, this study found that more than 1/3 of the older population experienced falls. Including functional mobility and balance training for social support programmes for older adults will be useful in helping them maintain physical health and improve their HRQOL.

Second, to improve their HRQOL, individualised policies to promote mental health for each age group should be implemented. It is necessary to establish social safety nets, such as individualised mental health follow-up systems for older adults living alone at public health centres or social support centres.

Third, older adults with low income were more likely to belong to the crisis type. Considering this, physical health, as well as social support and physical environment, should be improved for low-income individuals.

Fourth, this study confirmed an interactive relationship between physical and mental health in the older population. Although it is important to care for weaknesses in their health, interventions that maintain and promote healthy aspects are also crucial.

Fifth, older adults are expected to display a change in HRQOL over time. Therefore, identifying the variables that determine the stability/variability of the HRQOL over time will help improve the overall quality of life among older adults.

## 5. Conclusions

This study is a secondary analysis of a community health survey of G province from 2019 that was conducted to confirm latent classes of HRQOL in Korean older adults over 65 years of age and to analyse the characteristics of participants in each class and factors influencing each class.

Four latent classes of HRQOL were confirmed: stable, physical disability, emotional disability, and crisis. The present study was a cross-sectional study that utilised secondary data. Therefore, it was difficult to estimate the causality between variables. Moreover, since one area was selected as the sample, it was difficult to generalise the findings. Despite these limitations, the present study is significant as it confirmed latent classes of HRQOL in older adults and assessed their predictors.

Based on the findings of the present study, training on functional mobility and balance to prevent falls in older populations and individualised programmes to promote their mental health should be provided. Moreover, policies should increase medical accessibility and provide social support for those with a low income. Additionally, since physical, psychological, and social health in older adults are in an interactive relationship, a comprehensive care plan is needed to improve their HRQOL.

## Figures and Tables

**Figure 1 ijerph-18-07874-f001:**
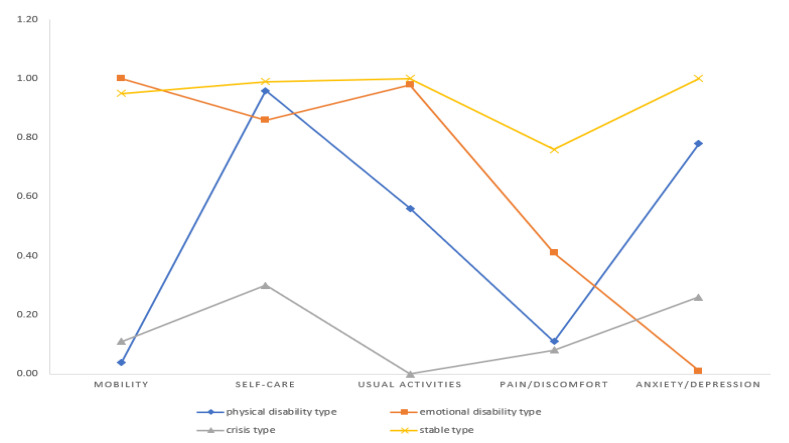
Latent classes of health-related quality of life.

**Table 1 ijerph-18-07874-t001:** Participants’ characteristics.

	Characteristics	Categories	N	%
General characteristics	Age group	65–70	3213	35.6
		71–75	2232	24.7
		76–80	1965	21.8
		≥ 81	1617	17.9
		M(SD)	74.07 (6.59)	
	Sex	Male	3892	43.1
		Female	5135	56.9
	Eligibility for basic livelihood security	Yes	559	6.2
		No	8468	93.8
	Household type	One-person household	1692	18.7
		Married couple household	4145	45.9
		Single parent household	175	1.9
		Others	3015	33.4
Physical health	Diagnosis of hypertension		5152	57.1
	Diagnosis of diabetes		1110	12.3
	Falls		0.32 (0.81)	
Mental health	Depression	Low or below	8558	94.8
		Moderate or higher	469	5.2
	Discomfort from cognitive decline		1.44 (0.90)	
	Happiness		6.66 (1.88)	
Social health	Trust in social environment		5.06 (1.63)	
	Degree of contact		3.46 (1.37)	
	Participation in social activities	Yes	5532	61.3
		No	3495	38.7

**Table 2 ijerph-18-07874-t002:** Fit indices of latent class analysis and distribution rate of HRQOLs (EQ-5D).

Number of Groups	AIC	BIC	saBIC	LMR	BLRT	Estimated Probability for Trajectory Group (%)					
						1	2	3	4	5	6
1	51,283.984	51,319.524	51,303.635	n/a	n/a	100.0					
2	41,886.226	41,964.414	41,929.457	<0.001	<0.001	66.8	33.2				
3	41,151.446	41,272.282	41,218.259	<0.001	<0.001	52.0	23.3	24.7			
4	41,013.431	41,176.914	41,103.824	<0.001	<0.001	57.3	16.6	10.1	16.0		
5	40,994.111	41,200.243	41,108.086	<0.001	<0.001	47.5	28.0	2.2	16.0	6.3	
6	41,002.081	41,250.860	41,139.636	0.733	1.000	40.6	10.5	2.9	22.6	10.0	13.4

Abbreviations: AIC = Akaike information criterion; BIC = Bayesian information criterion; saBIC = sample size adjusted BIC; LMR = Lo–Mendell–Rubin likelihood ratio test; BLRT = bootstrap likelihood ratio test.

**Table 3 ijerph-18-07874-t003:** Differences in indices of health-related quality of life among latent classes.

Group Indices	Class 1		Class 2		Class 3		Class 4		Total		F	*p*
	(*N* = 5173, 57.3%)		(*N* = 1501, 16.6%)		(*N* = 907, 10.0%)		(*N* = 1446, 16.0%)		(*N* = 9027, 100.0%)			
	M	(SD)	M	(SD)	M	(SD)	M	(SD)	M	(SD)		
Mobility	0.95	(0.21)	0.04	(0.20)	1.00	(0.00)	0.11	(0.31)	0.67	(0.47)	10,911.692	<0.001
Self-care	0.99	(0.10)	0.96	(0.20)	0.86	(0.35)	0.30	(0.46)	0.86	(0.35)	3272.555	<0.001
Usual activities	1.00	(0.07)	0.56	(0.50)	0.98	(0.15)	0.00	(0.00)	0.76	(0.43)	8937.071	<0.001
Pain/discomfort	0.76	(0.43)	0.11	(0.31)	0.41	(0.49)	0.08	(0.27)	0.51	(0.50)	1788.401	<0.001
Anxiety/depression	1.00	(0.00)	0.78	(0.41)	0.01	(0.20)	0.26	(0.44)	0.75	(0.43)	59.5.925	<0.001

**Table 4 ijerph-18-07874-t004:** Factors influencing latent classes.

	Characteristics	Categories	Comparison Group (Ref = Class 1)					
			Class 2		Class 3		Class 4	
			OR	*p*	OR	*p*	OR	*p*
General characteristics	Low income (ref = no)	Yes	1.18	0.245	1.08	0.641	2.04	<0.001
	Household type (ref = others)	One-person household	1.14	0.150	1.40	0.003	0.98	0.846
		Married couple household	0.84	0.019	1.06	0.542	0.94	0.489
		Single parent household	0.96	0.828	1.22	0.476	0.87	0.581
Physical health	Hypertension (ref = no)	Yes	1.30	<0.001	1.06	0.489	1.12	0.140
	Diabetes (ref = no)	Yes	1.11	0.281	1.05	0.658	0.99	0.916
	Number of falls		1.30	<0.001	1.26	<0.001	1.56	<0.001
Psychological health	Depression (ref = no)	Yes	2.67	<0.001	4.91	<0.001	10.30	<0.001
	Discomfort from cognitive decline		1.47	<0.001	1.79	<0.001	2.12	<0.001
	Happiness		0.87	<0.001	0.64	<0.001	0.65	<0.001
Social health	Degree of contact		0.92	<0.001	0.95	0.055	0.91	<0.001
	Trust in social environment		1.09	0.001	0.90	0.001	1.00	0.956
	Participation in social activities (ref = yes)	No	1.39	<0.001	0.89	0.172	1.82	<0.001

Controlled variable: Age, sex. ※ class 1 = stable type, class 2 = physical disability type, class 3 = emotional disability type, class 4 = crisis type.

## Data Availability

The datasets either generated, analysed, or both, during the current study are available in the Korea Centers for Disease Control and Prevention (KCDC) repository, accessed date (10 May 2021) https://chs.kdca.go.kr/chs/index.do.

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
