# Peer review of "A Latent Class Analysis of Health-Related Quality of Life in Korean Older Adults"

_ijerph, 2021, doi:10.3390/ijerph18157874_

Round 1

Reviewer 1 Report

This is an interesting article presenting data after analysis of health-related quality of life in older adults, minor revisions are suggested before possible publication.

A revision should be performed in the “Abstract” and “Materials and Methods” section; more information should be added in order to better understand the Korean administrative organisation (Paragraphs 2.1, 2.2 and 2.3); for example in the manuscript there are references to “G and K province” and “Basic Livelihood Security”, taking for granted  knowledge by the reader.

The fact that in physical health evaluation only hypertension, diabetes, and number of falls were analysed (without, for example, diagnosis of cancer) should be clearly highlighted as a limitation in the discussion section, given that there is a paragraph regarding “various chronic diseases” (lines 291-292).

Line 73 does not make sense without a verb. Please revise.

Line 340: year reference between brackets should be removed.

Table 1: wrong symbol for partecipants over 81 years of age.

A reference citing the works reporting PHQ-9 depression test questionnaire and Quality of life (EQ-5D) should be added.

Moreover, a revision of the references should be performed, as some articles have DOI while others do not.

Author Response

Thank you for your helpful comments on this manuscript. We tried to address the issues that you pointed out in your comments as much as possible, and the revised parts of the manuscript are shown in red font in the Word file. In addition, point-by-point responses to your comments are as follows.

Reviewer 2 Report

Manuscript ID: ijerph-1287256

The manuscript entitled “A latent class analysis of health-related quality of life in Korean
older adults” aimed to confirm latent classes in health-related quality of life in older adults and investigate the characteristics of participants in each class. Additionally, it was aimed to provide basic data to develop interventions for each quality-of-life class by analysing the predictors of each class.

  1. The sample is huge and allows sufficient statistical power to go into
    more detail. Did the pattern of variables predicting the HRQOS classes
    vary by sex, age, socioeconomic position, health measures, geographic
    region, etc. in terms of a moderation effect? It is important to
    identify the people at highest risk and most unfavorable patterns.

    2. Can the HRQOL classes identified be assumed to be stable over time?
    Which variables may determine this stability / variability over time?

    3. Besides the practical implications, the role for conceptual models
    may be elaborated in more detail. Can similar patterns be assumed for
    related outcomes such as well-being and health?

  1. Please rewrite the specific objectives of the study in the introduction section: pp2 L: 72-75.

  1. Do participants sign an informed consent to participate in this study?
  2. Can the author clarify the Inclusion criteria and exclusion criteria used?
  3. For depression, the PHQ-9 depression test questionnaire was used. Was this instrument being validate for the Korean population? Please provide references to the instrument.
  4. EQ-5D - was this instrument being validate for the Korean population? Please provide references to the instrument.
  5. Social health assessment. Was this instrument being validate for Korean population? Please provide references to the instrument.
  6. More information about the data collection is needed: the team who collected the data; Training of this team; general condition of data collection; etc
  7. Data Analysis require a better explanation.

Author Response

(The authors gave the same response as above.)

Reviewer 3 Report

This is an interesting study and I appreciated the opportunity to review it. I have some comments that I believe will strengthen the quality and impact of this manuscript.

The study refers to Korean society. I think the authors need to clarify whether they mean South and/or North Korea.

There does not seem to be an overall structure or flow to the Introduction. Instead the Introduction reads as a series of separate but not linked paragraphs and points. This needs to be addressed.

I thought the rationale for using latent class analysis and the description of LCA were both poor and need substantially improving and refining. I suggest reviewing published latent class analyses for template examples to copy.

I did not understand ‘The target sample size was 900 per public health centre.’

As someone not familiar with Korea, I would appreciate it if you describe what K Province is.

The internal consistency of any (multi-item) measures (eg PHQ9) used needs to be reported, as well as a brief overview of the published psychometric evidence for that measure (with references).

The LCA appear to have been performed and reported competently although the reporting could be clearer. There need to be references for the model fit criteria used to determined model selection. Again, I suggest reviewing published latent class analyses for template examples to follow in terms of how to report the Method and Results sections of an LCA study.

It is probably sufficient to report the OR results to 2 decimal places.

If longitudinal data are available, why not conduct a latent transition analysis.

I think the Discussion could be shortened.

The standard of English needs some improvement throughout.

Author Response

(The authors gave the same response as above.)

Reviewer 4 Report

Conclusively, I suggest this article can be accepted after a minor revision. This manuscript has two minor problems:

  1. The authors have reviewed some papers in the Discussion section. For example, the authors wrote, "Lee and Ko [25] reported....". I suggest this review may be put in the Introduction section.
  2. Authors may re-check the format of some journal references. Please unify this format. Some journals' names are italic; whereas, some others are not.

Author Response

(The authors gave the same response as above.)

Round 2

Reviewer 2 Report

I don't have further comments in the manuscript.